# On the Adversarial Robustness of Learning-based Conformal Novelty Detection

## Abstract

This paper studies the adversarial robustness of conformal novelty detection. In particular, we focus on AdaDetect, a powerful learning-based framework for novelty detection with finite-sample false discovery rate (FDR) control. While AdaDetect provides rigorous statistical guarantees under benign conditions, its behavior under adversarial perturbations remains unexplored. We first formulate an oracle attack setting that quantifies the worst-case degradation of FDR, deriving an upper bound that characterizes the statistical cost of attacks. This idealized formulation directly motivates a practical and effective attack scheme that only requires query access to AdaDetect's output labels. Coupling these formulations with two popular and complementary black-box adversarial algorithms, we systematically evaluate the vulnerability of AdaDetect on synthetic and real-world datasets. Our results show that adversarial perturbations can significantly increase the FDR while maintaining high detection power, exposing fundamental limitations of current error-controlled novelty detection methods and motivating the development of more robust alternatives.

## 1 Introduction

Consider the problem setup where training samples of *typical* events are collected and used for detecting abnormal events from a large number of testing samples, where the *abnormal* events follow a different distribution from the typical ones. This problem, known as *novelty detection* (e.g., Blanchard et al. (2010)), has attracted much attention recently through the lens of conformal p-values Vovk et al. (2005); Shafer & Vovk (2008); Bates et al. (2023); Mary & Roquain (2022); Liang et al. (2022); Marandon et al. (2024); Bashari et al. (2023). The underlying metric is the false discovery rate (FDR) Benjamini & Hochberg (1995); Benjamini & Yekutieli (2001); Efron et al. (2001); Genovese & Wasserman (2002); Storey (2002) that quantifies the false positives in a large-scale test dataset. Several frameworks have been developed with provable FDR control, requiring only exchangeability of the data under the null hypothesis. In particular, the ingenious method called *AdaDetect* Marandon et al. (2024) introduces an adaptive transformation of detection scores, learned from both null and alternative samples, that yields finite-sample FDR guarantees under exchangeability. AdaDetect can be viewed as an important extension of several existing approaches, including Bag of Null Statistics (BONus) Yang et al. (2021) and FDR control using conformal p-values Bates et al. (2023) (see (Marandon et al., 2024, Section 1.2) for detailed comparisons among existing approaches).

The appealing properties of AdaDetect, including the strong theoretical guarantees and efficient algorithm design, make it a potential strategy to empower existing safety-critical systems where training data samples are highly secure. For instance, in the banking system, customers' past personal transaction histories are highly protected yet can be used to enable fraud detection of new suspicious transactions when customers' account information is leaked and leveraged by malicious attackers. Thus, it is of great importance to quantify and evaluate the robustness of AdaDetect under various adversarial settings. In this work, we study the robustness of AdaDetect through the lens of adversarial machine learning that concerns the vulnerability of modern classifiers and detection systems under carefully designed perturbations. Importantly, we are interested in adversarial attacks *directly on the data*, rather than on transformed scores (e.g., p-values) — our study is the first of its kind in the literature, explicitly analyzing adversarial attacks on novelty detection systems while quantifying the cost in FDR control. Our proposed approach is flexible to incorporate existing adversarial machine learning attack algorithms. We hope to address the following two natural questions.

*How does a malicious attacker design an adversarial attack under FDR control?* As a first step in this direction, we focus on the setup where the test data can be attacked while keeping the training data intact, capturing the characteristics of safety-critical systems mentioned above. Specifically, we propose to first study the worst-case setting as a

baseline to quantify the loss in FDR under the strongest possible attack. This formulation and the corresponding analysis provide critical guidelines for the design of practical attack schemes. Interestingly, we have developed a heuristic yet powerful algorithm that almost achieves the worst possible attack in our synthetic and real data experiments.

*How to incorporate existing adversarial machine learning algorithms?* We propose a general methodology to make existing attack algorithms more effective. As a proof of concept, we adopt two popular off-the-shelf black-box adversarial attack algorithms, HopSkipJump Chen et al. (2020) and Boundary Attack Brendel et al. (2018), which only require the predicted label for refined data perturbation (see a detailed discussion in Section 3.1). These algorithms allow us to effectively change the score for the decision-making while minimizing the required changes in data. This approach enables directly attacks on the raw data, which is much more realistic than attacking scores of the data after some fixed transformation (e.g., p-values). For instance, there are works that focus on attacking p-values directly in the widely used Benjamini-Hochberg (BH) procedure for FDR control: a distributed setting using the BH procedure in the presence of compromised Byzantines Zhang et al. (2025), and on the adversarial robustness of the BH procedure through perturbation of p-values Chen et al. (2024).

## 1.1 CONTRIBUTIONS AND OUTLINE

The main contributions of this work are threefold. First, we introduce and design the adversarial attacks of the AdaDetect framework by proposing an oracle setting, when the attacker has access to both the model and test data labels, to quantify the upper bound on the loss in FDR (Section 3.1). Second, our oracle setting naturally motivates the design of a practical query-based attack scheme (detailed in Section 3.3), called the surrogate decision-based attack, where the attacker can only query for the labels of the test data. Third, in Section 4, the vulnerability of the AdaDetect scheme under our proposed attack strategies is extensively evaluated using two popular and complementary adversarial machine learning attack algorithms: HopSkipJump and Boundary Attack.

## 1.2 RELATED WORKS

**Novelty and anomaly detection with error control.** A growing body of work investigates conformal inference for novelty and anomaly detection with rigorous statistical guarantees (Laxhammar & Falkman, 2015; Smith et al., 2015; Ishimtsev et al., 2017; Guan & Tibshirani, 2022; Cai & Koutsoukos, 2020; Haroush et al., 2021). While classical detectors (Khan & Madden, 2014; Agrawal & Agrawal, 2015; Chalapathy & Chawla, 2019) often lack mechanisms for quantifying uncertainty, recent approaches provide explicit false discovery rate (FDR) control (Yang et al., 2021; Bates et al., 2023; Marandon et al., 2024). Among these, AdaDetect (Marandon et al., 2024) employs conformal $p$-values to guarantee FDR control while simultaneously learning the alternative distribution. Building on this line, Bashari et al. (2023) introduce a conformal $e$-value framework that derandomizes novelty detection and achieves rigorous FDR guarantees. AutoMS (Zhang et al., 2022) addresses model selection for out-of-distribution detection under controlled false discoveries, whereas online FDR-controlled anomaly detection (Rebjock et al., 2021) extends these guarantees to time-series data. These developments bring principled error control to novelty detection, but generally assume benign environments. Adversarial robustness in this context remains underexplored: Lo et al. (2022) shows that one-class detectors are vulnerable to adversarial manipulation, yet without statistical error guarantees. This gap highlights the need to integrate FDR-controlled detection with robustness analysis against adversarial threats.

**Adversarial machine learning.** Research on adversarial machine learning has revealed diverse classes of attacks depending on the adversary's knowledge and resources. In the **white-box** setting, adversaries have full knowledge of the model, including parameters and gradients. Early gradient-based attacks include FGSM Goodfellow et al. (2015), BIM/I-FGSM Kurakin et al. (2017), PGD Madry et al. (2018), DeepFool Moosavi-Dezfooli et al. (2016) and the CW attack Carlini & Wagner (2017). JSMA Papernot et al. (2016a) reduces perturbations to only a few critical dimensions. Universal and generative attacks extend beyond instance-specific perturbations (Moosavi-Dezfooli et al., 2017; Baluja & Fischer, 2017). More recent work considers spatial and semantic transformations, including Robust Physical Perturbations ($RP_2$) Eykholt et al. (2018) and spatially transformed adversarial examples Xiao et al. (2018). In the **black-box** setting, adversaries lack direct access to model gradients or parameters. Instead, they rely on querying the model or leveraging transferability. Score-based attacks estimate gradients using output probabilities, e.g., Zeroth-Order Optimization (ZOO) Chen et al. (2017), Natural Evolution Strategies (NES) Ilyas et al. (2018), and One-Pixel Attack Su et al. (2019). Decision-based attacks, such as the Boundary Attack Brendel et al. (2018), HopSkipJump Chen et al. (2020), and Sign-OPT Cheng et al. (2020), require only the final predicted label and progressively refine perturbations. Transfer-based methods exploit the phenomenon that adversarial examples often transfer across models: perturbations crafted on a surrogate can fool the target Papernot et al. (2016a). See Zheng et al. (2025)

for a comprehensive benchmark of black-box adversarial attacks. In response to the growing body of adversarial attacks, researchers have developed a range of defense mechanisms Xu et al. (2018); Madry et al. (2018); Zhang et al. (2019); Cohen et al. (2019); Papernot et al. (2016b); Wong & Kolter (2018); Metzen et al. (2017).

## 2 BACKGROUND

We have $n$ null training samples $\{Z_i\}_{i=1}^n$ sharing a common yet *unknown* marginal distribution $P_0$, and $m$ unlabeled testing samples $\{Z_i\}_{i=n+1}^{m+n}$ where $m_0$ of the testing samples share the same distribution as $P_0$ while the rest $m_1 = m - m_0$ of them follow different distributions. Let $\mathcal{H}_0$ contain all true null indices, while $\mathcal{H}_1$ contains all non-null indices in the testing data. We define the key performance metrics as follows. Let $V$ be the number of true nulls that are incorrectly rejected (false discoveries) and $R$ be the total number of rejections. The FDR is defined as

$$\text{FDR} = \mathbb{E}\left[\frac{V}{R \vee 1}\right], \tag{1}$$

where $R \vee 1 := \max\{R, 1\}$. The **power** measures the detection performance of non-nulls, defined as power $= \mathbb{E}[(R-V)/(m_1 \vee 1)]$, where $m_1 = |\mathcal{H}_1|$ is the number of non-nulls in the test set.

### 2.1 THE ADADETECT SCHEME MARANDON ET AL. (2024)

We use $\{X_1^{\text{train}}, \ldots, X_n^{\text{train}}\}$ to represent null training samples and $\{X_1^{\text{test}}, \ldots, X_m^{\text{test}}\}$ for the unlabeled testing samples. Following the notation from Marandon et al. (2024), we combine them into $\{Z_i\}_{i=1}^{n+m}$ in this work where $\{Z_i\}_{i=1}^n$ represent $\{X_1^{\text{train}}, \ldots, X_n^{\text{train}}\}$ and $\{Z_i\}_{i=n}^{n+m}$ represent $\{X_1^{\text{test}}, \ldots, X_m^{\text{test}}\}$. Regarding the data generation mechanism, we make the following general assumption, which is the same as (Marandon et al., 2024, Assumption 1).

**Assumption 1** (Exchangeability of nulls given non-nulls)**.**

$$(Z_1, \ldots, Z_n, Z_{n+i} : i \in \mathcal{H}_0) \mid (Z_{n+j} : j \in \mathcal{H}_1) \overset{d}{=} (Z_{\pi(1)}, \ldots, Z_{\pi(n)}, Z_{\pi(n+i)} : i \in \mathcal{H}_0) \mid (Z_{n+j} : j \in \mathcal{H}_1)$$

for any permutation $\pi$ of the indices $\{1, \ldots, n + m_0\}$.

AdaDetect is an adaptive novelty detection procedure that combines data-driven learning with distribution-free inference to provide finite-sample FDR control. The AdaDetect method partitions the null sample into training data $\{Z_i\}_{i=1}^k$ and calibration data $\{Z_i\}_{i=k+1}^n$. The algorithm proceeds as follows.

**Step 1: Learn score function.** Partition the data into the training data $\{Z_i\}_{i=1}^k$ and the mixed sample $\{Z_i\}_{i=k+1}^{n+m}$, which contains both calibration samples from $P_0$ and unlabeled test samples. Under the positive-unlabeled (PU) learning framework, apply a machine learning algorithm to learn a data-driven and measurable score function: $s : \mathcal{Z} \times \mathcal{Z}^k \times \mathcal{Z}^{n+m-k} \to \mathbb{R}$ as

$$s(z) := s(z; (Z_1, \ldots, Z_k), (Z_{k+1}, \ldots, Z_{n+m})) \tag{2}$$

satisfying the following permutation invariance property. For any permutation $\pi$ of $\{k+1, \ldots, n+m\}$,

$$s(z; (z_1, \ldots, z_k), (z_{\pi(k+1)}, \ldots, z_{\pi(n+m)})) = s(z; (z_1, \ldots, z_k), (z_{k+1}, \ldots, z_{n+m})). \tag{3}$$

**Step 2: Transform to scores.** Apply the learned function to obtain univariate scores

$$O_i = s(Z_i; (Z_1, \ldots, Z_k), (Z_{k+1}, \ldots, Z_{n+m})), \quad i \in [k+1 : n+m], \tag{4}$$

where a larger score indicates a higher likelihood of being a novelty.

**Step 3: Compute p-values.** For each test observation $Z_j$ with $j \in [n+1 : n+m]$, generate empirical p-values by comparing against the calibration set:

$$p_j = \frac{1}{n-k+1}\left(1 + \sum_{i=k+1}^n \mathbf{1}\{O_i > O_{n+j}\}\right), \quad j \in [1 : m]. \tag{5}$$

**Step 4: Apply BH procedure.** Apply the Benjamini-Hochberg algorithm to $(p_1, \ldots, p_m)$ to get the BH threshold $\tau$ at target level $\alpha$ and reject those p-values less than this threshold.

## 3  WORST-CASE ATTACK AND PRACTICAL ATTACK

In this section, we wish to study two attack schemes: an oracle attack and a query-based practical attack. Both schemes are compatible with existing black-box decision-based adversarial machine learning algorithms.

### 3.1  ORACLE SETTING: WORST-CASE ATTACK SCHEME

We first introduce the oracle setting to obtain an upper bound on the FDR loss when given the full information, enabling a theoretical analysis of the FDR behavior. Specifically, we assume that the attacker has access to the full dataset with correct labels, as well as all the configurations of the algorithm used in AdaDetect by the user. Specifically, the attacker has

> **Data:** Training samples $\{Z_j\}_{j=1}^{n}$ and test samples $\{Z_j\}_{j=n+1}^{m+n}$, and *the attacker knows which test samples are nulls and non-nulls*;
>
> **Algorithm:** All the information about the AdaDetect implemented by the user, including the machine learning model for the score function and its parameters.

We start by describing our first attack scheme (Step 1 and Step 2) and the outputs after applying AdaDetect directly on the attacked data (Step 3).

> **Step 1: Attack set selection.** Select a subset $\{Z_{n+i} : i \in \mathcal{A}\}$ from the true null test samples $\{Z_{n+i} : i \in \mathcal{H}_0\}$ as the attack target. We set the attack size as **fixed size** where $|\mathcal{A}| = m_a$ for some fixed number $m_a$.
>
> **Step 2: Decision-based adversarial perturbation.** Since the attacker has correct labels for all the data, the attacker can use them to train a score function $g(z)$ for the attack algorithm. We form the labeled dataset
>
> $$\mathcal{D}_{\text{oracle}} = \{(Z_i, Y_i^*)\}_{i=1}^{m+n}$$
>
> where $Y_i^* = 0$ for $i \in \mathcal{H}_0$, $Y_i^* = 1$ for $i \in \mathcal{H}_1$ and then train a score function
>
> $$g(z) \leftarrow \text{TrainScoreFunction}(\mathcal{D}_{\text{oracle}}).$$
>
> For each $i \in \mathcal{A}$, generate
>
> $$\widetilde{Z}_{n+i} = f_{\text{attack}}(Z_{n+i}; g(z)) \tag{6}$$
> $$:= f_{\text{attack}}(Z_{n+i}; \{Z_1, ..., Z_n, Z_{n+j} : j \in \mathcal{H}_0 \setminus \mathcal{A}\}, (Z_{n+j} : j \in A \cup \mathcal{H}_1)) \tag{7}$$
>
> such that $\mathbf{1}\{g(Z_{n+i}) \geq 0.5\} \neq \mathbf{1}\{g(\widetilde{Z}_{n+i}) \geq 0.5\}$, meaning that the decision is altered. Here we write
>
> $$\{Z_1, ..., Z_n, Z_{n+j} : j \in \mathcal{H}_0 \setminus \mathcal{A}\}$$
>
> as an *unordered* set to highlight that $f_{\text{attack}}$ does not depend on the order of elements in this set.
>
> **Step 3: Applying AdaDetect on the attacked data.** After the attack, the user applies AdaDetect and computes the score function as the first step. As the data is now changed by the attacker, we denote the score function after the attack by $\tilde{s}$, and the empirical $p$-values after the attack by $\tilde{p}_i$ for $i \in [1 : m]$. We stress that $\tilde{s}$ still satisfies equation 3.

The key in this oracle setting is that the attacker knows which ones are true nulls in the test data. The attacker will simply pick $\mathcal{A}$ with $m_a = |\mathcal{A}|$, which consists of a *fixed* set of indices of nulls in the test data (i.e., there is no randomness in $\mathcal{A}$ and $m_a$). Our proposed methodology is flexible in that it can incorporate existing adversarial machine learning attack algorithms.

The following proposition is critical for our analysis. It holds since (I) $f_{\text{attack}}(\cdot\,; g(z))$ only relies on the score function $g(z)$, and (II) $g(z)$ is invariant to order of elements in $\{Z_1, \ldots, Z_{n+i} : i \in \mathcal{H}_0 \setminus \mathcal{A}\}$ as they are all labeled as 0.

**Proposition 1.** *$f_{\text{attack}}$ does not depend on the order of elements in $\{Z_1, ..., Z_n, Z_{n+j} : j \in \mathcal{H}_0 \setminus \mathcal{A}\}$.*

In Section 4, we evaluate adversarial robustness using two representative decision-based attacks: the Boundary Attack (Brendel et al., 2018) and the HopSkipJumpAttack (HSJA) (Chen et al., 2020). Decision-based attacks, which rely

solely on query access to the decision function, capture adversarial capabilities more realistically than white-box or score-based methods. Moreover, evaluating FDR loss under such attacks provides a stringent robustness assessment, as these "blind" perturbations often induce more severe failures than those observed under other threat models. The two algorithms were chosen for their complementary properties. The Boundary Attack is a seminal decision-based approach that operates via a random-walk strategy, starting from an adversarial example and progressively reducing the perturbation while remaining misclassified. It is conceptually simple, model-agnostic, and widely adopted as a baseline in the literature. In contrast, HSJA is a more recent attack that achieves state-of-the-art query efficiency by combining binary search with adaptive estimation of the decision boundary's normal vector. While both attacks require only hard-label access to the model, Boundary Attack provides a robust baseline, whereas HSJA represents a stronger and more query-efficient adversary. Together, they allow us to assess robustness against both classical and modern decision-based adversarial paradigms.

The attack in the last step pushes the score above the decision boundary. It is important to note that the output of the HSJA scheme Chen et al. (2020), i.e., $f_{attack}$ = HSJA indeed does not depend on the order of elements in $\{Z_1, ..., Z_n, Z_{n+i} : i \in \mathcal{H}_0 \setminus \mathcal{A}\}$. The same holds for the Boundary Attack (Brendel et al., 2018).

**Remark 1.** *Although HSJA is sometimes described as a gradient estimation method, it does not require differentiability of the model; instead, it approximates the boundary's normal vector using only hard-label queries. This makes it applicable even to non-differentiable classifiers such as random forests.*

### 3.2 ANALYSIS

In our oracle setting, we denote the corresponding FDR as $\text{FDR}^*_{\text{attack}}$. Our main theorem quantifies the loss in FDR caused by the attack. The proof is deferred to Appendix C.

**Theorem 1.** *Consider that $\mathcal{A}$ is a fixed set of indices with $m_a = |\mathcal{A}|$. Under Assumption 1, with the score function $\tilde{s}$ satisfying the permutation invariance property in equation 3 and the attack scheme $f_{attack}$ being order-invariant as in equation 7, the FDR after the attack is*

$$FDR^*_{attack} \leq \alpha + m_a \cdot \mathbb{E}\left[\frac{1}{\widetilde{R} \vee 1}\right], \tag{8}$$

*where the expectations are taken over the randomness in the training and test samples $\{Z_j\}_{j=1}^{m+n}$.*

Unlike the original AdaDetect, which guarantees FDR $\leq \alpha$ in benign settings, Theorem 1 provides an upper-bound under adversarial perturbations. In our proofs, we start with decomposing the FDR into attacked and unattacked components, and our key technical innovations (Lemmas 1 and 2) say that the first term $\mathbb{E}[\sum_{i \in \mathcal{H}_0 \setminus \mathcal{A}} \frac{\widetilde{V}_i}{\widetilde{R} \vee 1}]$ remains bounded by $\alpha$ even after perturbations, demonstrating that AdaDetect's statistical control over unattacked samples is preserved.

**Remark 2.** *It turns out that the proof techniques for this oracle setting in Theorem 1 can be adapted to less stringent settings where the true labels of test samples are unknown to the attacker. We report them in Appendix D. In the next section, we propose a heuristic algorithm that is motivated by our oracle setting.*

The key lemma below shows the conditional exchangeability we need for Theorem 1. First, we introduce the following notation for simplicity of presentation. Denote the number of unattacked true null test samples by $\tilde{m}_0$. In order to simplify notation, let

$$U_{\setminus \mathcal{A}} = (U_1, \ldots, U_{n+\tilde{m}_0}) := (Z_1, \ldots, Z_n, Z_{n+i} : i \in \mathcal{H}_0 \setminus \mathcal{A}),$$
$$U_{\mathcal{A}} = (U_{n+\tilde{m}_0+1}, \ldots, U_{n+m_0}) := (Z_{n+i} : i \in \mathcal{A}),$$
$$\widetilde{U}_{\mathcal{A}} = (\widetilde{U}_{n+\tilde{m}_0+1}, \ldots, \widetilde{U}_{n+m_0}) := (\widetilde{Z}_{n+i} : i \in \mathcal{A}),$$
$$V = (V_1, \ldots, V_{m_1}) := (Z_{n+i} : i \in \mathcal{H}_1).$$

With a slight abuse of notation, the condition equation 7 on $f_{\text{attack}}$ can be simplified as

$$\widetilde{Z}_{n+i} = f_{\text{attack}}(Z_{n+i} \,; \, U_{\setminus \mathcal{A}}, U_{\mathcal{A}} \cup V), \tag{9}$$

where we stress that $f_{\text{attack}}$ does not depend on the order of the elements in $U_{\setminus \mathcal{A}}$. The following lemma establishes a crucial property not addressed in the original AdaDetect:

*A form of conditional exchangeability is preserved even after adversarial perturbations.*

The key insight is that by conditioning on not only non-null data $V$ but also the attack outcomes $\widetilde{U}_\mathcal{A}$, the unattacked null samples maintain their exchangeable structure. See the proof in Appendix A.

**Lemma 1.** *Under the setting of Theorem 1, we have*

$$(U_{k+1}, \ldots, U_{n+\tilde{m}_0}) \mid V \cup \widetilde{U}_\mathcal{A} \stackrel{d}{=} (U_{\pi(k+1)}, \ldots, U_{\pi(n+\tilde{m}_0)}) \mid V \cup \widetilde{U}_\mathcal{A}$$

*for any permutation $\pi$ of the indices $\{k+1, \ldots, n+\tilde{m}_0\}$.*

**Remark 3.** *For our theoretical analysis, we find it is sufficient to establish exchangeability for the unattacked true null elements indexed from $k+1$ onwards to derive the FDR upper bound. While the original AdaDetect analysis demonstrates exchangeability for all true null elements including the first $k$ training samples, this broader exchangeability does not hold in our adversarial setting due to the attack's dependence on the complete dataset. Specifically, their exchangeability expression covers $(Z_1, \ldots, Z_n, Z_{n+i} : i \in \mathcal{H}_0)$ conditioned on $V$, while our restricted result only requires $(Z_{k+1}, \ldots, Z_n, Z_{n+i} : i \in \mathcal{H}_0 \setminus \mathcal{A})$ to be exchangeable conditioned on $\widetilde{U}_\mathcal{A}$ and $V$.*

Now we show that the conditional exchangeability of data in Lemma 1 can be carried over to the scores. See the proof in Appendix B.

**Lemma 2.** *Under the setting of Lemma 1 and assume that $\tilde{s}$ satisfies equation 2, then we have*

$$(\tilde{s}(U_{k+1}), \ldots, \tilde{s}(U_{n+\tilde{m}_0})) \mid (\tilde{s}(Z_{n+j}) : j \in \mathcal{H}_1) \cup (\tilde{s}(\widetilde{Z}_{n+j}) : j \in \mathcal{A})$$

$$\stackrel{d}{=} (\tilde{s}(U_{\pi(k+1)}), \ldots, \tilde{s}(U_{\pi(n+\tilde{m}_0)})) \mid (\tilde{s}(Z_{n+j}) : j \in \mathcal{H}_1) \cup (\tilde{s}(\widetilde{Z}_{n+j}) : j \in \mathcal{A})$$

*for any permutation $\pi$ of the indices $\{k+1, \ldots, n+\tilde{m}_0\}$.*

Building on Lemma 1 and Lemma 2, the following result follows directly from (Marandon et al., 2024, Theorem A.1 (iii) and (iv))) and we skip the proof.

**Lemma 3.** *Under the Lemma 2 setting. Let $i \in \mathcal{H}_0$ be an unattacked null index and $S_i := \tilde{s}(\widetilde{Z}_i)$ for $i \in [1 : m+n]$. Define*

$$\widetilde{W}_i = \{S_{k+1}, \ldots, S_n, S_{n+i}\} \cup (S_{n+j} : j \neq i, j \in \mathcal{H}_0) \cup (S_{n+j} : j \in \mathcal{H}_1). \tag{10}$$

*Then we have*

*(i) The $p$-value $\tilde{p}_i$ is independent of $\widetilde{W}_i$.*

*(ii) The quantity $(n - k + 1)\tilde{p}_i$ is uniformly distributed on the integers $\{1, \ldots, n-k+1\}$.*

### 3.3 SURROGATE DECISION-BASED ATTACK SCHEME

Motivated by our oracle setting, we consider a practical scenario, where the attacker does not have the training samples $\{Z_j\}_{j=1}^n$ and does not know the true labels of the test samples, but is allowed to query the label from the user who applies AdaDetect, with underlying machine learning algorithms unknown to the attacker. Such query access is a standard assumption in adversarial settings as mentioned in Section 1.2, reflecting a realistic constraint on the attacker's capability. Specifically, the attack has

> **Data:** Only test samples $\{Z_j\}_{j=n+1}^{m+n}$, but the attacker does not know which ones are nulls and non-nulls;

> **Query:** The attacker can query the user (who owns all the training and test data and the AdaDetect algorithm) to obtain the labels for all the testing data $\{Z_j\}_{j=n+1}^{m+n}$.

We propose a *surrogate score function* $g(z)$, trained on the pseudo-labeled dataset $\mathcal{D} = \{(Z_{n+i}, y_i)\}_{i=1}^m$, where we refer to $\{y_i\}_{i=1}^m$ as pseudo-labels since , unlike the true labels available in our oracle setting, these are the labels assigned by AdaDetect on the entire test set. This surrogate score function approximates AdaDetect's binary decision: $\mathbf{1}\{g(Z_{n+i}) \geq 0.5\} \approx y_i$, enabling decision-based adversarial attacks on a black-box detector.

> **Step 1: Initial detection.** The attacker queries the labels of the test data from the user. Upon request, the user applies AdaDetect to the full test set $\{Z_{n+i}\}_{i=1}^m$ at once, producing pseudo-labels $Y_i \in \{0, 1\}$ where,

$$(Y_1, \ldots, Y_m) = \text{AdaDetect}\left(\{Z_{n+i}\}_{i=1}^m\right).$$

**Step 2: Surrogate score function training.** Assuming AdaDetect has reasonable detection power, we form the pseudo-labeled dataset $\mathcal{D} = \{(Z_{n+i}, Y_i)\}_{i=1}^m$ and train a surrogate score function

$$g(z) \leftarrow \text{TrainScoreFunction}(\mathcal{D}).$$

**Step 3: Attack set selection and adversarial perturbation.** Select a subset $\mathcal{A}$ from the unrejected test samples as the attack target, where $|\mathcal{A}| = m_{\mathcal{A}} = \lfloor \gamma(m - R) \rfloor$ with $\gamma \in (0, 1]$ being the attack intensity parameter. For each $i \in \mathcal{A}$, compute

$$\widetilde{Z}_{n+i} = f_{\text{attack}}(Z_{n+i}, g(z)).$$

It is important to note that our surrogate decision-based attack does not require information about the algorithm, making it a practical attack scheme. This point is also highlighted in our experiment section through mismatched setups where the user and attacker adopt two different algorithms to learn the score function (see Experiment 3 and Experiment A.1 for details).

**Remark 4.** *We note that another possible attack is to treat AdaDetect as a black-box and apply a decision-based attack directly to its outputs. However, this might require a prohibitively large query budget, since changing the empirical FDR demands perturbing many test samples, and each sample in turn requires multiple queries to attack successfully.*

## 4 EXPERIMENTS

As explained in Section 3.1, we will focus on two adversarial machine learning attacks: HSJA and Boundary attack. Each experiment is repeated 20 times to calculate the mean and variance of the FDR and power. We denote the AdaDetect score function by $s(z)$ and the score function used for attack by $g(z)$. The estimated upper bound according to equation 8 is computed by $\alpha + m_a \cdot \frac{1}{20} \sum_{i=1}^{20} \left( \frac{1}{\widehat{R}^{(i)} \vee 1} \right)$. This is computed for all of the rest of the experiments and they are listed in the tables.

### 4.1 SYNTHETIC DATA EXPERIMENTS

We conduct four comprehensive experiments to evaluate our method's performance across different data distributions and model configurations. All experiments use the following base parameters: training sample size $n = 5000$, testing sample size $m = 1000$, $k = 4000$, true null data $m_0 = 900$, and significance level $\alpha = 0.1$. We test our approach on three types of synthetic data distributions: independent Gaussian, non-Gaussian, and exchangeable Gaussian data. *Due to space limitations, we defer the synthetic data generation and one more synthetic experiments in Appendix E.*

**Experiment 1: Varying attack size with random forest (RF) models.** We evaluate the impact of attack size on both oracle and surrogate attack performance using identical RF architectures for both score functions. The attack size $m_a$ sweeps between 50 and 200. We assess how each attack scheme performs under different attack scales with homogeneous RF configurations.

We only tested HSJA in synthetic data experiments while both of HSJA and Boundary attack will be evaluated in real-world data experiments. The result shows that the FDR for both attack schemes are under the theoretical upper bound, and oracle attack outperforms surrogate attack in non-gaussian case when the original power is low. Overall, both attack schemes successfully increase the FDR across varying attack sizes.

### 4.2 REAL-WORLD DATA EXPERIMENTS

We evaluated the attack performance on four real-world datasets with diverse characteristics and application domains. All experiments use the same base parameters ($n$, $m$, $k$, $m_0$, $\alpha$) as the synthetic data experiments. Each experiment is repeated 20 times to calculate the mean and variance of the FDR and power. We consider four popular datasets: Shuttle, Credit Card, KDDCup99, and Mammography (see data descriptions and one more experiment in Appendix E due to space limitations).

**Experiment 2: Real-world data with RF models.** We evaluated both oracle and surrogate attack performance in the five real-world datasets using identical RF architectures for both score functions, with attack size $m_a = 200$. This

Table 1: Experiment 1: FDR + RF

| Dataset | Independent Gaussian | Non-Gaussian | Exchangeable Gaussian |
|---|---|---|---|
| original FDR | $0.08 \pm 0.03$ | $0.08 \pm 0.04$ | $0.08 \pm 0.04$ |
| oracle ($m_a = 50$) | $0.36 \pm 0.02$ | $0.40 \pm 0.05$ | $0.38 \pm 0.02$ |
| surrogate ($m_a = 50$) | $0.34 \pm 0.02$ | $0.20 \pm 0.05$ | $0.37 \pm 0.02$ |
| estimated upper bound | 0.43 | 0.42 | 0.41 |
| oracle ($m_a = 200$) | $0.67 \pm 0.00$ | $0.71 \pm 0.01$ | $0.69 \pm 0.00$ |
| surrogate ($m_a = 200$) | $0.67 \pm 0.00$ | $0.64 \pm 0.01$ | $0.67 \pm 0.00$ |
| estimated upper bound | 0.80 | 0.78 | 0.77 |

Table 2: Experiment 1: Power + RF

| Dataset | Independent Gaussian | Non-Gaussian | Exchangeable Gaussian |
|---|---|---|---|
| original power | $0.96 \pm 0.02$ | $0.55 \pm 0.06$ | $1.00 \pm 0.00$ |
| oracle ($m_a = 50$) | $0.99 \pm 0.01$ | $0.87 \pm 0.01$ | $1.00 \pm 0.00$ |
| surrogate ($m_a = 50$) | $0.96 \pm 0.02$ | $0.65 \pm 0.05$ | $1.00 \pm 0.00$ |
| oracle ($m_a = 200$) | $0.99 \pm 0.01$ | $0.98 \pm 0.01$ | $1.00 \pm 0.00$ |
| surrogate ($m_a = 200$) | $0.96 \pm 0.02$ | $0.78 \pm 0.05$ | $1.00 \pm 0.00$ |

allows us to compare attack effectiveness across different real-world data characteristics. We compare the boundary attack with our default HSJA under both oracle and surrogate attack schemes, using identical RF architectures for both score functions with $m_a = 200$. The result shows that our oracle and surrogate attack schemes and decision-based algorithms (HopSkipJump and Boundary) can significantly increase the FDR in comparison to the original FDR.

Table 3: Experiment 2: FDR + RF

| Dataset | Credit-card | Shuttle | KDD | Mammography |
|---|---|---|---|---|
| original FDR | $0.08 \pm 0.03$ | $0.01 \pm 0.00$ | $0.04 \pm 0.02$ | $0.04 \pm 0.08$ |
| oracle+**hop.** | $0.60 \pm 0.02$ | $0.65 \pm 0.02$ | $0.48 \pm 0.10$ | $0.51 \pm 0.10$ |
| surrogate+**hop.** | $0.56 \pm 0.02$ | $0.66 \pm 0.03$ | $0.45 \pm 0.08$ | $0.45 \pm 0.11$ |
| oracle+**bound.** | $0.61 \pm 0.02$ | $0.68 \pm 0.02$ | $0.65 \pm 0.07$ | $0.61 \pm 0.05$ |
| surrogate+**bound.** | $0.64 \pm 0.03$ | $0.70 \pm 0.02$ | $0.67 \pm 0.06$ | $0.57 \pm 0.04$ |
| estimated upper bound | 0.85 | 0.73 | 0.69 | 0.88 |

We make two important observations. Firstly, compared to the other three datasets, the original power on Mammography is relatively low ($\sim 0.48$). As a consequence, the surrogate method learns the score function from less accurate labels, leading to a larger gap in FDR between the oracle vs. the surrogate. Similar phenomena also appear in the next real-world experiments, as well as the two in the Appendix E. Secondly, the power after attack often increases, since the attack targets data points near the decision boundary in practice, some of which belong to the alternative hypothesis and are thus more likely to be correctly rejected.

Table 4: Experiment 2: Power + RF

| Dataset | Credit-card | Shuttle | KDD | Mammography |
|---|---|---|---|---|
| original power | $0.78 \pm 0.03$ | $0.84 \pm 0.02$ | $0.88 \pm 0.04$ | $0.48 \pm 0.09$ |
| oracle+**hop.** | $0.86 \pm 0.03$ | $0.99 \pm 0.01$ | $0.94 \pm 0.05$ | $0.67 \pm 0.07$ |
| surrogate+**hop.** | $0.87 \pm 0.03$ | $0.99 \pm 0.01$ | $0.93 \pm 0.05$ | $0.80 \pm 0.05$ |
| oracle+**bound.** | $0.98 \pm 0.02$ | $0.97 \pm 0.02$ | $0.95 \pm 0.01$ | $0.80 \pm 0.09$ |
| surrogate+**bound.** | $0.95 \pm 0.03$ | $0.96 \pm 0.03$ | $0.96 \pm 0.01$ | $0.78 \pm 0.10$ |

**Experiment 3: Real-world data with mismatched configurations.** We apply four mismatched score function configurations (RF–NN, RF–RF) to the five real-world datasets with a fixed attack size of $m_a = 200$, evaluating both the oracle and surrogate attack performance. This enables us to assess how different model combinations affect each attack type's performance across various real-world scenarios. The results indicate that an attacker can employ a model different from the user's and still inflate the FDR beyond the target level. Our experiments show that using the neural network configuration as the attacker's model can substantially speed up the attack process. The results for RF-RF configuration is covered in Experiment 2.

Table 5: Experiment 3: FDR + RF-NN

| Dataset | Credit-card | Shuttle | KDD | Mammography |
|---|---|---|---|---|
| original FDR | $0.09 \pm 0.05$ | $0.01 \pm 0.01$ | $0.02 \pm 0.01$ | $0.09 \pm 0.05$ |
| oracle+**bound.** | $0.64 \pm 0.03$ | $0.69 \pm 0.02$ | $0.69 \pm 0.02$ | $0.69 \pm 0.01$ |
| surrogate+**bound.** | $0.60 \pm 0.02$ | $0.50 \pm 0.03$ | $0.67 \pm 0.03$ | $0.64 \pm 0.01$ |
| estimated upper bound | 0.79 | 0.72 | 0.69 | 0.85 |

## 5 DISCUSSION

We believe that this work opens up a wide range of possible directions concerning the interplay between adversarial robustness and conformal novelty detection. We briefly comment on two potential research directions.

**Defense and robust training.** In response to the growing body of research on adversarial attacks, researchers have developed a range of defense mechanisms. Early approaches focused on input preprocessing, such as feature squeezing Xu et al. (2018) or randomized transformations, but these were often circumvented by adaptive adversaries. More principled methods emphasize robust training. Adversarial training Madry et al. (2018) has become the de facto standard, where models are trained on adversarial examples generated during training to improve robustness. In the context of novelty detection with FDR control, these techniques suggest potential defenses against adversarially induced FDR inflation: robust training can make the decision boundary less susceptible to small perturbations, while randomized smoothing could stabilize conformal scores or p-values, thereby preserving statistical error guarantees under attack. Exploring such defenses offers a promising direction for integrating adversarial robustness with principled error control.

**Attack on training or calibration data.** As a first step in understanding the robustness of AdaDetect, we consider the security-critical scenarios where the training data is highly secure. It would be interesting to study the impact of attacks on the null samples, including the training and calibration data. This can be a suitable setup for less powerful agents, such as power-limited sensors or local servers in decentralized formulations (e.g. Zhang et al. (2025); Pournaderi & Xiang (2023)). For instance, consider that each sensor is deployed in the environment for monitoring, then attacking the calibration data is more reasonable and powerful, as it changes the reference for all the test samples.

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

# A  APPENDIX: PROOF OF LEMMA 1

*Proof.* From Assumption 1, for any permutation $\pi$ of the indices $\{1, \ldots, n + \tilde{m}_0\}$ such that $\pi(i) = i$ for $i \leq k$, we have

$$(U_{\setminus \mathcal{A}} \mid V) \stackrel{d}{=} (U_{\setminus \mathcal{A}}^{\pi} \mid V). \tag{11}$$

From the property of the attack algorithm, we have for each $i \in \mathcal{A}$ that

$$\widetilde{Z}_{n+i} = f_{\text{attack}}(Z_{n+i} \,;\, U_{\setminus \mathcal{A}}, U_{\mathcal{A}} \cup V), \tag{12}$$

and we write this in a compact form as $\widetilde{U}_{\mathcal{A}} = f_{\text{attack}}(U_{\mathcal{A}} \,;\, U_{\setminus \mathcal{A}}, U_{\mathcal{A}} \cup V)$. We want to show that

$$(U_{\setminus \mathcal{A}} \mid f_{\text{attack}}(U_{\mathcal{A}} \,;\, U_{\setminus \mathcal{A}}, U_{\mathcal{A}} \cup V) \stackrel{d}{=} (U^{\pi} \mid f_{\text{attack}}(U_{\mathcal{A}} \,;\, U_{\setminus \mathcal{A}}^{\pi}, U_{\mathcal{A}} \cup V). \tag{13}$$

According to Proposition 1, $f_{\text{attack}}$ does not depend on order of the elements from $U_{k+1}$ to $U_{n+\tilde{m}_0}$ in $U_{\setminus \mathcal{A}}$, so we have $\widetilde{U}_{\mathcal{A}} = f_{\text{attack}}(U_{\mathcal{A}} \,;\, U_{\setminus \mathcal{A}}^{\pi}, U_{\mathcal{A}} \cup V)$. For any measurable set $\mathcal{U}_{\setminus \mathcal{A}}$ in the support of $U_{\setminus \mathcal{A}}$, we have

$$\mathrm{P}\left(U_{\setminus \mathcal{A}} \in \mathcal{U}_{\setminus \mathcal{A}} \mid \widetilde{U}_{\mathcal{A}} = \tilde{u}_{\mathcal{A}}, V = v\right) = \frac{\mathrm{P}(U_{\setminus \mathcal{A}} \in \mathcal{U}_{\setminus \mathcal{A}}, \, f_{\text{attack}}(U_{\mathcal{A}} \,;\, U_{\setminus \mathcal{A}}, U_{\mathcal{A}} \cup V) = \tilde{u}_{\mathcal{A}}, \, V = v)}{\mathrm{P}(f_{\text{attack}}(U_{\mathcal{A}} \,;\, U_{\setminus \mathcal{A}}, U_{\mathcal{A}} \cup V) = \tilde{u}_{\mathcal{A}}, \, V = v)}. \tag{14}$$

According to Assumption 1, we know that all the elements inside $U_{\setminus \mathcal{A}}$ are exchangeable given $V$. Along with the assumption that $f_{\text{attack}}$ satisfies equation 7, we have that

$$\mathrm{P}(U_{\setminus \mathcal{A}} \in \mathcal{U}_{\setminus \mathcal{A}}, \, f_{\text{attack}}(U_{\mathcal{A}} \,;\, U_{\setminus \mathcal{A}}, U_{\mathcal{A}} \cup V) = \tilde{u}_{\mathcal{A}} \mid V = v) \tag{15}$$

$$= \mathrm{P}(U_{\setminus \mathcal{A}}^{\pi} \in \mathcal{U}_{\setminus \mathcal{A}}, \, f_{\text{attack}}(U_{\mathcal{A}} \,;\, U_{\setminus \mathcal{A}}^{\pi}, U_{\mathcal{A}} \cup V) = \tilde{u}_{\mathcal{A}} \mid V = v). \tag{16}$$

Therefore,

$$\mathrm{P}\left(U_{\setminus \mathcal{A}} \in \mathcal{U}_{\setminus \mathcal{A}} \mid \widetilde{U}_{\mathcal{A}} = \tilde{u}_{\mathcal{A}}, V = v\right) = \mathrm{P}\left(U_{\setminus \mathcal{A}}^{\pi} \in \mathcal{U}_{\setminus \mathcal{A}} \mid \widetilde{U}_{\mathcal{A}} = \tilde{u}_{\mathcal{A}}, V = v\right), \tag{17}$$

i.e., $\{U_{k+1}, \ldots, U_{n+\tilde{m}_0}\}$ is conditionally exchangeable given $(\widetilde{U}_{\mathcal{A}}, V) = (f_{\text{attack}}(U_{\mathcal{A}} \,;\, U_{\setminus \mathcal{A}}, U_{\mathcal{A}} \cup V), V)$. $\square$

# B  APPENDIX: PROOF OF LEMMA 2

*Proof.* Let

$$Q = h(U, \widetilde{U}_{\mathcal{A}}, V) = ((Z_1, \ldots, Z_k), \{\widetilde{Z}_{n+i} : i \in \mathcal{A} \cup Z_{n+i} : i \in \mathcal{H}_1\}). \tag{18}$$

By Lemma 1, we have

$$(U, \widetilde{U}_{\mathcal{A}}, V) \stackrel{d}{=} (U^{\pi}, \widetilde{U}_{\mathcal{A}}, V). \tag{19}$$

This implies

$$(U, \widetilde{U}_{\mathcal{A}}, V, Q) = (U, \widetilde{U}_{\mathcal{A}}, V, h(U, \widetilde{U}_{\mathcal{A}}, V)) \stackrel{d}{=} (U^{\pi}, \widetilde{U}_{\mathcal{A}}, V, h(U^{\pi}, \widetilde{U}_{\mathcal{A}}, V)). \tag{20}$$

Since the permutation keeps those indices fixed, by the definition of $Q$, we have $h(U^{\pi}, \widetilde{U}_{\mathcal{A}}, V) = h(U, \widetilde{U}_{\mathcal{A}}, V) = Q$. Thus

$$(U, \widetilde{U}_{\mathcal{A}}, V, Q) \stackrel{d}{=} (U^{\pi}, \widetilde{U}_{\mathcal{A}}, V, Q). \tag{21}$$

Applying the score function $s$ to each $U_i$, we obtain

$$(S_1, \ldots, S_{n+\tilde{m}_0}) \mid \widetilde{U}_{\mathcal{A}}, V, Q \stackrel{d}{=} (S_{\pi(1)}, \ldots, S_{\pi(n+\tilde{m}_0)}) \mid \widetilde{U}_{\mathcal{A}}, V, Q. \tag{22}$$

Since $\pi(i) = i$ for all $i \leq k$ and $i > n + \tilde{m}_0$, the permutation only affects the scores

$$((S_{k+1}, \ldots, S_n), S_{n+i} : i \in \mathcal{H}_0 \setminus \mathcal{A}).$$

Therefore, Eq. equation 22 implies

$$(S_{k+1}, \ldots, S_n), S_{n+i} : i \in \mathcal{H}_0 \setminus \mathcal{A}) \mid \widetilde{U}_{\mathcal{A}}, V, Q$$

$$\stackrel{d}{=} (S_{\pi(k+1)}, \ldots, S_{\pi(n)}), S_{\pi(n+i)}) : i \in \mathcal{H}_0 \setminus \mathcal{A}) \mid \widetilde{U}_{\mathcal{A}}, V, Q.$$

Now, we justify why conditioning on $(\widetilde{U}_{\mathcal{A}}, V, Q)$ is equivalent to conditioning only on $(\tilde{s}(Z_{n+j}) : j \in \mathcal{H}_1) \cup (\tilde{s}(\widetilde{Z}_{n+j}) : j \in \mathcal{A})$. First, note that $Q = h(U, \widetilde{U}_{\mathcal{A}}, V)$ is a deterministic function of $U$, $\widetilde{U}_{\mathcal{A}}$ and $V$. The score function $s$ depends on $\widetilde{U}_{\mathcal{A}}$ and $V$ only through this transformation $Q$.

Furthermore, those conditional scores are the results of a deterministic function of $(Q, \widetilde{U}_{\mathcal{A}}, V)$. Therefore, the scores $(\tilde{s}(Z_{n+j}) : j \in \mathcal{H}_1 \cup \mathcal{A})$ are fully determined once $\widetilde{U}_{\mathcal{A}}$, $V$ and $Q$ are fixed, and vice versa.

As a result, conditioning on $(\widetilde{U}_{\mathcal{A}}, V, Q)$ is equivalent to conditioning on the non-null and attacked scores, thus

$$(\tilde{s}(Z_{k+1}), \dots, \tilde{s}(Z_n), \tilde{s}(Z_{n+i}) : i \in \mathcal{H}_0 \setminus \mathcal{A}) \text{ is exchangeable conditioned on}$$
$$(\tilde{s}(Z_{n+j}) : j \in \mathcal{H}_1) \cup (\tilde{s}(\widetilde{Z}_{n+j}) : j \in \mathcal{A})$$

$\square$

## C APPENDIX: PROOF OF THEOREM 1

*Proof of Theorem 1.* Let $\widetilde{V}_i$ denote the indicator function for the rejection of hypothesis $i$ and $\tilde{\tau}$ be the BH threshold under the adversarial attack, where

$$\widetilde{V}_i = \mathbf{1}\{\tilde{p}_i \leq \alpha\,(\tilde{\tau}/m)\}. \tag{23}$$

We can decompose $\text{FDR}_{\text{attack}}$ as follows,

$$\text{FDR}_{\text{attack}} = \mathbb{E}\left[\sum_{i \in \mathcal{H}_0 \setminus \mathcal{A}} \frac{\widetilde{V}_i}{\widetilde{R} \vee 1}\right] + \mathbb{E}\left[\sum_{i \in \mathcal{A} \cap \mathcal{H}_0} \frac{\widetilde{V}_i}{\widetilde{R} \vee 1}\right]. \tag{24}$$

Let $\widetilde{R} = \sum_{i=1}^m \widetilde{V}_i$ be the total number of rejections after the attack. For the second term, we can bound it by

$$\mathbb{E}\left[\sum_{i \in \mathcal{A} \cap \mathcal{H}_0} \frac{\widetilde{V}_i}{\widetilde{R} \vee 1}\right] \leq \mathbb{E}\left[\frac{|\mathcal{A} \cap \mathcal{H}_0|}{\widetilde{R} \vee 1}\right] \overset{(a)}{=} \mathbb{E}\left[\frac{m_{\mathcal{A}}}{\widetilde{R} \vee 1}\right], \tag{25}$$

where $(a)$ follows since $\mathcal{A} \subseteq \mathcal{H}_0$. For the first term, define $S_i = \tilde{s}(\widetilde{Z}_i)$ for $i \in [1 : m+n]$. Fix any $i \in \mathcal{H}_0 \setminus \mathcal{A}$, and for $j \neq i$:

$$C_{i,j} = \frac{1}{n-k+1}\left(\sum_{s \in \{S_{k+1}, \dots, S_n, S_{n+i}\}} \mathbf{1}\{s > S_{n+j}\}\right). \tag{26}$$

Define the empirical $p$-values after attack

$$\tilde{p}_i = \frac{1 + \sum_{j=k+1}^n \mathbf{1}\{\tilde{s}(Z_j) \geq \tilde{s}(\widetilde{Z}_{n+i})\}}{n-k+1}.$$

We now create the *auxiliary* p-value vector $(p'_1, \dots, p'_m)$ by

$$p'_j = \begin{cases} \dfrac{1}{n-k+1} & \text{if } j = i, \\ C_{i,j} & \text{if } j \neq i. \end{cases} \tag{27}$$

Hence $p'_j \leq \tilde{p}_j$ whenever $\tilde{p}_j \leq \tilde{p}_i$ is guaranteed by construction ($C_{i,j}$ is smaller or equal if $\tilde{p}_j \leq \tilde{p}_i$), and $p'_j = \tilde{p}_j$ if $\tilde{p}_j > \tilde{p}_i$. Condition (63) in (Marandon et al., 2024, Lemma D.6) is satisfied for *all* $j \neq i$. Recall $\tilde{\tau}$ is the BH index for $(\tilde{p}_1, \dots, \tilde{p}_m)$ and let $\tau'_i := \tau'_{\text{BH}}$ be the BH index for $(p'_1, \dots, p'_m)$. By (Marandon et al., 2024, Lemma D.6), we obtain

$$\left\{\tilde{p}_i \leq \alpha\left(\tfrac{\tilde{\tau}}{m}\right)\right\} = \left\{\tilde{p}_i \leq \alpha\left(\tfrac{\tau'_i}{m}\right)\right\} \subseteq \{\tilde{\tau} = \tau'_i\}. \tag{28}$$

Focusing on $i \in \mathcal{H}_0 \setminus \mathcal{A}$, we have

$$\mathbf{1}\{\tilde{p}_i \leq \alpha\,(\tilde{\tau}/m)\} = \mathbf{1}\{\tilde{p}_i \leq \alpha\,(\tau'_i/m)\}. \tag{29}$$

Summing over $i \in \mathcal{H}_0 \setminus \mathcal{A}$,

$$\mathbb{E}\left[\sum_{i \in \mathcal{H}_0 \setminus \mathcal{A}} \frac{\widetilde{V}_i}{\widetilde{R} \vee 1}\right] = \mathbb{E}\left[\sum_{i \in \mathcal{H}_0 \setminus \mathcal{A}} \frac{\mathbf{1}\{\tilde{p}_i \leq \alpha\left(\tilde{\tau}/m\right)\}}{\tilde{\tau}}\right] = \mathbb{E}\left[\sum_{i \in \mathcal{H}_0 \setminus \mathcal{A}} \frac{\mathbf{1}\{\tilde{p}_i \leq \alpha\left(\tau_i'/m\right)\}}{\tau_i'}\right]. \tag{30}$$

We define

$$\widetilde{W}_i = \{S_{k+1}, \ldots, S_n, S_{n+i}\} \cup (S_{n+j} : j \neq i, j \in H_0) \cup (S_{n+j} : j \in H_1), \tag{31}$$

so $\tau_i'$ is $\widetilde{W}_i$-*measurable*. Hence

$$\mathbb{E}\left[\sum_{i \in \mathcal{H}_0 \setminus \mathcal{A}} \frac{\widetilde{V}_i}{\widetilde{R} \vee 1}\right] = \mathbb{E}\left[\sum_{i \in \mathcal{H}_0 \setminus \mathcal{A}} \mathbb{E}\left[\frac{\mathbf{1}\left\{\tilde{p}_i \leq \alpha\left(\tau_i'/m\right)\right\}}{\tau_i'} \,\bigg|\, \widetilde{W}_i\right]\right] \tag{32}$$

$$= \mathbb{E}\left[\sum_{i \in \mathcal{H}_0 \setminus \mathcal{A}} \frac{1}{\tau_i'} \mathbb{E}\left[\mathbf{1}\left\{\tilde{p}_i \leq \alpha\left(\tau_i'/m\right)\right\} \,\bigg|\, \widetilde{W}_i\right]\right], \tag{33}$$

where the last equality is due to $\tau_i'$ acting as a known constant inside that inner conditional. From Lemma 3, we know that $(n - k + 1)\tilde{p}_i$ is exactly the rank of $S_{n+i}$ among $\{S_{k+1}, \ldots, S_n, S_{n+i}\}$ and $\tilde{p}_i$ is independent of $\widetilde{W}_i$. As a result, $(n - k + 1)\tilde{p}_i$ is uniform on $\{1, \ldots, n - k + 1\}$, independent of $\tau_i'$. Thus

$$\mathbb{E}\left[\mathbf{1}\{\tilde{p}_i \leq \alpha\left(\tau_i'/m\right)\} \,\bigg|\, \widetilde{W}_i\right] = \mathbb{P}\left((n - k + 1)\tilde{p}_i \leq \alpha(n - k + 1)\frac{\tau_i'}{m} \,\bigg|\, \widetilde{W}_i\right) = \frac{\lfloor \alpha(n - k + 1)\frac{\tau_i'}{m}\rfloor}{n - k + 1}. \tag{34}$$

Hence

$$\mathbb{E}\left[\sum_{i \in \mathcal{H}_0 \setminus \mathcal{A}} \frac{\widetilde{V}_i}{\widetilde{R} \vee 1}\right] = \mathbb{E}\left[\sum_{i \in \mathcal{H}_0 \setminus \mathcal{A}} \frac{\lfloor \alpha(n - k + 1)\tau_i'/m\rfloor}{(n - k + 1)\tau_i'}\right] \leq m_0 \cdot \mathbb{E}\left[\frac{\lfloor \alpha(n - k + 1)\tau_i'/m\rfloor}{(n - k + 1)\tau_i'}\right]. \tag{35}$$

The first term is thus bounded by $\alpha$.

$\square$

## D   APPENDIX: DIRECT DECISION-BASED ATTACK SCHEME

Different from the oracle setting, we assume that the attacker has access to

   **Data:** Training samples $\{Z_j\}_{j=1}^n$ and test samples $\{Z_j\}_{j=n+1}^{m+n}$, but *the attacker does not know which test samples are nulls and non-nulls*;

   **Algorithm:** All the information about the AdaDetect implemented by the user, including the machine learning model for the score function and its parameters.

With such information at hand, the attacker is able to apply AdaDetect locally on $\{Z_j\}_{j=1}^{m+n}$, to obtain the score function $s(z)$ defined as in equation 2.

We start with describing our first attack scheme as follows.

   **Step 1: Initial detection and BH labeling.** Using the training data $\{Z_j\}_{j=1}^n$ and the mixed sample $\{Z_j\}_{j=k+1}^{n+m}$, form the dataset

$$\mathcal{D} = \{(Z_i, Y_i)\}_{i=1}^{n+m}$$

where $Y_i = 0$ for $i \in [1 : k]$ and $Y_i = 1$ for $i \in [k+1 : n+m]$ using the positive-unlabeled (PU) framework. Train the score function

$$s(z) \leftarrow \text{TrainScoreFunction}(\mathcal{D}).$$

Note that $s(z)$ automatically satisfies the condition in equation 3 as $Y_i$ for $i \in [k+1 : n+m]$ are the same. Compute empirical $p$-values for $i \in [1 : m]$

$$\hat{p}_i = \frac{1 + \sum_{j=k+1}^{n} \mathbf{1}\{s(Z_j) \geq s(Z_{n+i})\}}{n - k + 1}.$$

Then apply the BH-procedure to $(\hat{p}_1, \ldots, \hat{p}_m)$ to get BH threshold $\hat{\tau}$ at target level $\alpha$ and produce binary labels

$$(\hat{Y}_1, \ldots, \hat{Y}_m) = \text{BH}\left((\hat{p}_1, \ldots, \hat{p}_m), \alpha\right),$$

where $\hat{Y}_i = 1$ indicates rejection (detected as non-null) and $\hat{Y}_i = 0$ indicates non-rejection (undetected).

**Step 2: Attack set selection.** Within the set of test samples (i.e., with indices from $[n+1 : n+m]$), select a subset $\{Z_{n+i} : i \in \mathcal{A}\}$ from the unrejected test samples as the attack target.

We set the attack size as (1) **fixed size** where $|\mathcal{A}| = m_a$ for some fixed number $m_a$, or (2) **random size** where $m_{\mathcal{A}} = \lfloor \gamma(m - R) \rfloor$, where $\gamma \in (0, 1]$ is an "attack intensity" parameter specified by the attacker. One natural choice of $\mathcal{A}$ is to select the unrejected hypotheses with the smallest $p$-values (i.e., those closest to the rejection boundary). Let $(i_1, i_2, \ldots, i_{m-R})$ denote the indices of unrejected hypotheses ordered by their $p$-values: $\hat{p}_{i_1} \leq \hat{p}_{i_2} \leq \cdots \leq \hat{p}_{i_{m-R}}$. Then we define $\mathcal{A} = \{i_1, i_2, \ldots, i_{m_{\mathcal{A}}}\}$. This selects the $m_{\mathcal{A}}$ unrejected indices with the smallest $p$-values, targeting hypotheses that are close to $\hat{\tau}$ and making it an effective attack strategy as demonstrated in our experiments.

**Step 3: Decision-based adversarial perturbation.** For each $i \in \mathcal{A}$, generate

$$\widetilde{Z}_{n+i} = f_{\text{attack}}(Z_{n+i}; \ s(z)) \tag{36}$$
$$:= f_{\text{attack}}(Z_{n+i}; \ \{Z_{k+1}, ..., Z_n, Z_{n+j} : j \in \mathcal{H}_0 \setminus \mathcal{A}\}, (Z_{n+j} : j \in A \cup \mathcal{H}_1), (Z_1, \ldots, Z_k)) \tag{37}$$

such that $\mathbf{1}\{s(Z_{n+i}) \geq 0.5\} \neq \mathbf{1}\{s(\widetilde{Z}_{n+i}) \geq 0.5\}$, meaning that the decision is altered. Here we write

$$\{Z_{k+1}, ..., Z_n, Z_{n+j} : j \in \mathcal{H}_0 \setminus \mathcal{A}\}$$

as an *unordered* set to highlight that $f_{\text{attack}}$ does not depend on the order of elements in this set.

**Step 4: Applying AdaDetect on the attacked data.** After the attack, the user applies AdaDetect and computes the score function as the first step. As the data is now changed by the attacker, we denote the score function after the attack by $\tilde{s}$, and the empirical $p$-value after the attack by $\tilde{p}_i$ for $i \in [1 : m]$. We stress that $\tilde{s}$ still satisfies equation 3.

The attack set $\mathcal{A}$ is inherently random because it depends on the outcome of the BH procedure in Step 1, which in turn depends on the computed $p$-values of the random test samples $\{Z_{n+i}\}_{i=1}^{m}$. More specifically, each $p$-value $\hat{p}_i$ relies on the entire dataset, including both the training samples $\{Z_j\}_{j=1}^{n}$ and test samples $\{Z_{n+j}\}_{j=1}^{m}$, through the score function computation and ranking procedure. In other words, $\mathcal{A}$ is a complex yet deterministic function of the complete dataset.

**Proposition 2.** $f_{attack}$ *does not depend on the order of elements in* $\{Z_{k+1}, ..., Z_n, Z_{n+j} : j \in \mathcal{H}_0 \setminus \mathcal{A}\}$.

This proposition captures the main subtle yet important difference between this attack scheme and the oracle setting in Theorem 1. It holds because $f_{\text{attack}}(\cdot; \ s(z))$ only relies on the score function $s(z)$, and $s(z)$ is invariant to order of elements in $\{Z_{k+1}, \ldots, Z_{n+m}\}$ according to equation 3, as a consequence of the PU framework.

### D.1 ANALYSIS

In this setting, we denote the corresponding FDR as $\text{FDR}^*_{\text{attack–decision}}$, and quantify the loss in FDR caused by the attack.

**Theorem 2.** *Consider that $\mathcal{A}$ is a fixed set of indices with $m_a = |\mathcal{A}|$. Under Assumption 1, with the score function $\tilde{s}$ satisfying the permutation invariance property in equation 3 and the attack scheme $f_{attack}$ being order-invariant as in equation 37, the FDR after the attack is*

$$FDR^*_{attack–decision} \leq \alpha + m_a \cdot \mathbb{E}\left[\frac{1}{\widetilde{R} \vee 1}\right], \tag{38}$$

*where the expectations are taken over the randomness in the training and test samples $\{Z_j\}_{j=1}^{m+n}$.*

**Remark 5.** *It has been proved that AdaDetect has a strong detection power (the probability of correctly rejecting a non-null), as shown in (Marandon et al., 2024, Theorem 5.1). This implies that the set $\mathcal{A}$ will nearly contain all indices from true nulls because the non-nulls are mostly rejected. We use this to show that the upper bound $\mathbb{E}[\frac{m_{\mathcal{A}}}{R \vee 1}]$ in equation 1 is relatively tight (see details in Proposition 3).*

*Proof of Theorem 2.* It follows directly from proof of Theorem 1, as the only difference between Theorem 2 and Theorem 1 is that the score function is trained differently. But according to Proposition 2, the score function for attack is still invariant under the permutation of $\{Z_{k+1}, ..., Z_n, Z_{n+j} : j \in \mathcal{H}_0 \setminus \mathcal{A}\}$, which makes the rest of the proof exactly the same as that for Theorem 1. $\qquad\square$

**Corollary 1.** *Consider that $\mathcal{A}$ is random with fixed size $|\mathcal{A}| = m_a$. Under Assumption 1, with equation 3 and equation 7, the FDR after the attack is*

$$FDR_{attack} \leq FDR^*_{attack-decision} \leq \alpha + m_a \cdot \mathbb{E}\left[\frac{1}{\widetilde{R} \vee 1}\right], \tag{39}$$

*where the expectations are taken over the randomness in the training and test samples $\{Z_j\}_{j=1}^{m+n}$, which induces randomness in $\mathcal{A}$ and $\widetilde{R}$.*

Unlike the original AdaDetect, which guarantees FDR $\leq \alpha$ in benign settings, Corollary 1 provides an FDR upper-bound under adversarial perturbations. We also have the following corollary about the setting when $\mathcal{A}$ is random with a fixed size $m_{\mathcal{A}}$.

**Corollary 2.** *Consider that $\mathcal{A}$ is random with a random size $|\mathcal{A}| = m_{\mathcal{A}}$. Under Assumption 1, with equation 3 and equation 7, the FDR after the attack is*

$$FDR_{attack} \leq \alpha + \mathbb{E}\left[\frac{m_{\mathcal{A}}}{\widetilde{R} \vee 1}\right], \tag{40}$$

*where the expectations are taken over the randomness in the training and test samples $\{Z_j\}_{j=1}^{m+n}$, which induces randomness in $\mathcal{A}$ and $\widetilde{R}$.*

In the following proposition, we show that the upper bound in equation 25 is relatively tight by making a connection between step (a) in equation 25 and the power of AdaDetect. Roughly speaking, the upper bound is relatively tight when the power of AdaDetect is decent. Instead of showing detailed technical steps following the proof of (Marandon et al., 2024, Theorem 5.1), we choose to provide a high-level argument to connect the power of AdaDetect and our upper bound in equation 25.

**Proposition 3.** *Under the assumptions in (Marandon et al., 2024, Theorem 5.1) and when $\mathcal{A}$ is randomly selected from the unrejected indices with $m_{\mathcal{A}} = \lfloor \gamma(m - R) \rfloor$, we have that for some small $\delta'$ and $\eta'$,*

$$P(\mathcal{A} \subseteq \mathcal{H}_0) \geq (1 - \delta') \cdot l(\eta'), \tag{41}$$

*where $l(\eta') \to 1$ as $\eta' \to 0$.*

According to (Marandon et al., 2024, Theorem 5.1), we have that the rejection set by AdaDetect at level $\lambda$, denoted by AdaDetect$_\lambda$ satisfies

$$P\left(\frac{|\text{AdaDetect}_\lambda \cap \mathcal{H}_1|}{m_1} \geq 1 - \eta\right) \geq 1 - \delta, \tag{42}$$

for some small $\delta$ and $\eta$. This can be adapted to our setting as follows, with one approximation, where we treat the data samples being attacked as non-nulls. After the attack, we have

$$P\left(\frac{|\widetilde{\text{AdaDetect}}_\lambda \cap (\mathcal{H}_1 \cup \mathcal{A})|}{|\mathcal{H}_1 \cup \mathcal{A}|} \geq 1 - \eta'\right) \geq 1 - \delta', \tag{43}$$

where $\widetilde{\text{AdaDetect}}_\lambda$ denotes the rejection set after the attack, and we note that the number of non-nulls becomes $|\mathcal{H}_1 \cup \mathcal{A}| \leq m_1 + m_{\mathcal{A}}$ after the attack.

Since $\widetilde{\mathcal{R}} = \widetilde{\mathrm{AdaDetect}_\lambda}$ and $|\widetilde{\mathcal{R}} \cap (\mathcal{H}_1 \cup \mathcal{A})| = \widetilde{R} - \widetilde{V}$, we have that the unrejected set contains $(|\mathcal{H}_1 \cup \mathcal{A}| - (\widetilde{R} - \widetilde{V}))) \leq$ $|\mathcal{H}_1 \cup \mathcal{A}| \cdot \eta'$ with probability at least $1 - \delta'$. Define $\mathcal{E} = \left\{ |\widetilde{R} - \widetilde{V}| \geq |\mathcal{H}_1 \cup \mathcal{A}| \cdot (1 - \eta') \right\}$. We use the shorthand $Z := \{Z_i\}_{i=1}^{n+m}$ to denote the whole dataset. Given any fixed dataset $Z = z$, the only remaining randomness comes from the random selection (and it is independent of $Z$), while the random variables $\widetilde{R}$, $\mathcal{A}$, and $m_{\mathcal{A}}$ take on realizations as $\widetilde{R}(z)$, $\mathcal{A}(z)$, and $m_{\mathcal{A}(z)}$, respectively. We now have

$$P\left(\mathcal{A} \subseteq \mathcal{H}_0\right) \geq P(\mathcal{E}) \cdot P\left(\mathcal{A} \subseteq \mathcal{H}_0 \mid \mathcal{E}\right)$$

$$\geq (1 - \delta') \cdot \int_z \frac{\binom{m - \widetilde{R}(z) - |\mathcal{H}_1 \cup \mathcal{A}(z)| \cdot \eta'}{m_{\mathcal{A}(z)}}}{\binom{m - \widetilde{R}(z)}{m_{\mathcal{A}(z)}}} \cdot P(Z = z | \mathcal{E}) \, dz := (1 - \delta') \cdot l(\eta'),$$

where $l(\eta') \to 1$ as $\eta' \to 0$. The last step follows since given event $\mathcal{E}$ and $\{Z = z\}$, we have $|\widetilde{\mathcal{R}}(z) \cap (\mathcal{H}_1 \cup \mathcal{A}(z))| \geq |\mathcal{H}_1 \cup \mathcal{A}(z)| \cdot (1 - \eta')$, which implies the number of unrejected non-nulls is smaller than $|\mathcal{H}_1 \cup \mathcal{A}(z)| \cdot \eta'$.

# E  APPENDIX: ADDITIONAL EXPERIMENTS

**Synthetic data generation.** We generate two types of data: null samples from distribution $P_0$ and non-null samples from distribution $P_1$. We let $d = 20$ for all the synthetic data.

- **Independent Gaussian:** We consider
  $$P_0 = \mathcal{N}(0, I_d), \quad P_1 = \mathcal{N}(\mu, I_d),$$
  where $\mu \in \mathbb{R}^d$ is a sparse mean shift vector: the first five coordinates are set to $\sqrt{2 \log(d)}$ and the remaining coordinates are zero.

- **Non-Gaussian:** We let the first two coordinates of nulls and non-nulls be drawn independently from Beta distributions, with
  $$P_0 : \quad (X_1, X_2) \sim \mathrm{Beta}(5, 5), \qquad P_1 : \quad (X_1, X_2) \sim \mathrm{Beta}(1, 3).$$
  The remaining coordinates are drawn i.i.d. from $\mathrm{Beta}(1, 1)$ under both $P_0$ and $P_1$.

- **Exchangeable Gaussian:** Let $T = \mathcal{N}(\mu, \Sigma)$ be the $d$-variate Gaussian distribution with mean vector $\mu = [\mu_1, \ldots, \mu_d]^\top$ and covariance matrix $\Sigma = [\sigma_{ij}]_{i,j=1}^d$. Suppose $T$ is exchangeable, i.e.,
  $$\mu_i = \mu_j =: a, \qquad \sigma_{ii} = \sigma_{jj} =: b^2, \qquad \sigma_{ij} = \sigma_{kl} =: c,$$
  for all $i, j, k, l$ with $i \neq j$ and $k \neq l$. Then the covariance matrix can be written as
  $$\Sigma = c \mathbf{1} \mathbf{1}^\top + (b^2 - c) I_d,$$
  where $\mathbf{1} = [1, \ldots, 1]^\top \in \mathbb{R}^d$.
  We define the null and non-null distributions as
  $$P_0 = \mathcal{N}(a\mathbf{1}, \Sigma), \qquad P_1 = \mathcal{N}((a + \delta)\mathbf{1}, \Sigma),$$
  where $\delta > 0$ introduces a mean shift across all coordinates. Thus $P_0$ and $P_1$ share the same exchangeable covariance structure but differ in their mean vectors.

**Four real-world datasets.**

- **Shuttle:** Radiator data onboard space shuttles. Instances from class 1 are considered nominal, while instances from classes 2–7 are considered novelties Dua & Graff (2019).
- **Credit Card:** Transactions made by credit cards over two days, some of which are fraudulent Dal Pozzolo et al. (2015).
- **KDDCup99:** A set of network connections that includes a variety of simulated intrusions Stolfo et al. (1999).
- **Mammography:** Features extracted from mammograms, some with microcalcifications Woods et al. (1993).

**Experiment A.1: Mismatched score function configurations.** We investigate both oracle and surrogate attack performance when using different model architectures and parameters for the score functions, with attack size $m_a = 200$. This experiment comprises four distinct configurations for each attack type:

- **RF–NN:** AdaDetect score function $s(z)$ uses an RF, attack model score function $g(z)$ uses an NN.
- **RF–RF:** Both score functions use RFs with same hyperparameters.

Table 6: Experiment A.1: FDR + RF-NN

| Dataset | Independent Gaussian | Non-Gaussian | Exchangeable Gaussian |
|---|---|---|---|
| original FDR | $0.08 \pm 0.03$ | $0.08 \pm 0.04$ | $0.08 \pm 0.04$ |
| oracle ($m_a = 200$) | $0.66 \pm 0.00$ | $0.70 \pm 0.02$ | $0.67 \pm 0.00$ |
| surrogate ($m_a = 200$) | $0.68 \pm 0.00$ | $0.63 \pm 0.01$ | $0.65 \pm 0.02$ |
| estimated upper bound | 0.77 | 0.76 | 0.67 |

**Experiment A.2: Real-world data with NN models.** This experiment employs NN architectures for both score functions on the four real-world datasets, evaluating both oracle and surrogate attack performance with attack size $m_a = 200$. This allows us to evaluate how each attack type performs when both the target model and attacker use NN-based approaches on realistic data distributions. We compare boundary attack with our default HSJA attack under both oracle and surrogate attack schemes, using identical RF architectures for both score functions with $m_a = 200$. The result shows that both HSJA and boundary attack are successful at rasing the FDR for NN models in real-world data.

Table 7: Experiment A.2: FDR + NN

| Dataset | Credit-card | Shuttle | KDD | Mammography |
|---|---|---|---|---|
| original FDR | $0.09 \pm 0.05$ | $0.01 \pm 0.01$ | $0.02 \pm 0.01$ | $0.09 \pm 0.05$ |
| oracle+**hop.** | $0.67 \pm 0.04$ | $0.44 \pm 0.01$ | $0.59 \pm 0.03$ | $0.78 \pm 0.01$ |
| surrogate+**hop.** | $0.67 \pm 0.05$ | $0.43 \pm 0.00$ | $0.61 \pm 0.02$ | $0.65 \pm 0.02$ |
| oracle+**bound.** | $0.66 \pm 0.02$ | $0.36 \pm 0.05$ | $0.47 \pm 0.06$ | $0.64 \pm 0.05$ |
| surrogate+**bound.** | $0.65 \pm 0.02$ | $0.45 \pm 0.09$ | $0.43 \pm 0.05$ | $0.61 \pm 0.04$ |
| estimated upper bound | 0.77 | 0.76 | 0.81 | 0.80 |

Table 8: Experiment A.2: Power + NN

| Dataset | Credit-card | Shuttle | KDD | Mammography |
|---|---|---|---|---|
| original power | $0.80 \pm 0.03$ | $0.84 \pm 0.09$ | $0.78 \pm 0.04$ | $0.53 \pm 0.09$ |
| oracle+**hop.** | $0.95 \pm 0.03$ | $0.98 \pm 0.01$ | $0.88 \pm 0.02$ | $0.65 \pm 0.01$ |
| surrogate+**hop.** | $0.86 \pm 0.04$ | $0.99 \pm 0.01$ | $0.86 \pm 0.03$ | $0.87 \pm 0.01$ |
| oracle+**bound.** | $0.93 \pm 0.03$ | $0.94 \pm 0.01$ | $0.99 \pm 0.01$ | $0.77 \pm 0.06$ |
| surrogate+**bound.** | $0.95 \pm 0.02$ | $0.99 \pm 0.01$ | $0.97 \pm 0.01$ | $0.80 \pm 0.07$ |

