# OpenReview forum: "ON THE ADVERSARIAL ROBUSTNESS OF LEARNING-BASED CONFORMAL NOVELTY DETECTION"
_ICLR.cc/2026/Conference — ICLR 2026 Conference Withdrawn Submission_

### Official Review · Reviewer_wVpK · 2025-10-31

**Soundness:** 3
**Presentation:** 3
**Contribution:** 3
**Rating:** 6
**Confidence:** 3

**Summary:**

This paper is investigating the adversarial robustness of AdaDetect which is a conformal novelty detection framework with finite sample false discovery rate (FDR) control. The authors propose two attack schemes 1) an oracle attack (with full knowledge of data labels and model) and 2) a practical surrogate decision-based attack (requiring only query access to output labels). They derive theoretical upper bounds on FDR degradation and evaluate attacks using HopSkipJump and Boundary Attack algorithms on synthetic and real-world datasets.

**Strengths:**

- The paper is clear and easy to follow.
- It explores the novel setting of studying FDR-controlled novelty detection.
- The paper provides rigorous theoretical analysis, deriving formal upper bounds on FDR degradation, proving conditional exchangeability preservation, and establishing connections between attack effectiveness and AdaDetect's detection power.
- The proposed framework is evaluated on both synthetic and real-world datasets using multiple models.

**Weaknesses:**

- The paper does not explore sufficient variation in attack sizes. While it examines a fixed attack size m_a, it could have investigated adaptive strategies to strengthen the experiments. Additionally, the paper lacks experiments on different values of m_a and provides no discussion on the effects of varying this parameter.
- Given that experiments show oracle attacks sometimes significantly outperform surrogate attacks, the paper lacks discussion on when these gaps are large, what causes them, and whether it is possible to bound them theoretically.
- The hyperparameter values for Attack Intensity used in the experiments are unclear. Furthermore, the paper lacks additional experiments demonstrating how this hyperparameter affects FDR.

**Questions:**

1. Based on the Mammography results in Table 3, is it reasonable to conclude that the upper bound is not tight for cases with low detection power?
2. How many queries do HSJA and Boundary Attack require per sample to achieve the reported FDR increases? What's the tradeoff between query budget and attack effectiveness?
3. Can the defender detect ongoing attacks by monitoring query patterns or input distributions? What signals would indicate adversarial activity?
4. Will you release code for reproducibility?

---

### Official Review · Reviewer_C3w9 · 2025-10-31

**Soundness:** 2
**Presentation:** 2
**Contribution:** 1
**Rating:** 4
**Confidence:** 3

**Summary:**

The paper explores the adversarial robustness of conformal novelty detection, focusing on AdaDetect, which controls the FDR under exchangeability. The authors formalize an attack and prove an upper bound on the FDR under attack; propose a practical, surrogate, decision-based attack that requires only hard-label queries to AdaDetect, train a surrogate on pseudo-labels, select unrejected points near the threshold, then apply HSJA/Boundary; and empirically demonstrate sizable FDR inflation on synthetic and tabular real-world datasets (Shuttle, Credit Card, KDDCup99, Mammography).

**Strengths:**

Robust novelty detection is an important and timely problem.


The paper includes a theoretical analysis of FDR under attack.

**Weaknesses:**

[A] RODEO: Robust Outlier Detection via Exposing Adaptive Out-of-Distribution Samples

[B] Adversarially Robust Out-of-Distribution Detection Using Lyapunov-Stabilized Embeddings

[C] ATOM: Robustifying Out-of-distribution Detection Using Outlier Mining


W1) There are several papers on adversarially robust anomaly/novelty detection in computer vision [A, B, C]. Why aren’t these cited or discussed? Those works typically demonstrate that detectors are vulnerable and (propose defenses to improve robustness. Please clarify how your problem setup, threat model, and contributions differ from and improve upon [A–C].

W2) A major part of the paper appears to show that models are vulnerable to attacks and  design attacks for them. Since it is well known that most models are intrinsically vulnerable to adversarial perturbations, what is the core novelty here beyond confirming that fact in your setting? If I’m missing a key insight, please make it explicit.

W3) The paper does not clearly specify the distance metrics and constraints used per dataset (e.g., $L_2$​ bounds, feature-wise box constraints, integer/one-hot features) or whether off-manifold tabular perturbations are considered feasible.

W4)  For tabular data, how do you select an adversarial perturbation budget that preserves semantics and validity (e.g., monotonic relationships, categorical encodings, integrality, domain ranges)? In images, budgets are bounded by perceptual tolerances; for tabular data, a principled rationale or sensitivity analysis would help.

W5) The study is tightly focused on AdaDetect, raising concerns about generality and applicability. To what extent do your findings transfer to other setups?

**Questions:**

See Weaknesses.

---

### Official Review · Reviewer_4buF · 2025-11-01

**Soundness:** 2
**Presentation:** 2
**Contribution:** 1
**Rating:** 2
**Confidence:** 5

**Summary:**

In terms of motivation, while the authors acknowledge a growing body of recent work that incorporates conformal inference (e.g. Bates et al. 2023's conformal p-values) for novelty/anomaly detection, they believe that these methods'
 `Adversarial robustness $\ldots$ remains underexplored."
Thus, they motivate their line of work by claiming that AdaDetect (Marandon et al 2024) is an ``ingenious" method and a potential strategy to empower safety-critical systems, so that it is of great importance to quantify and evaluate the robustness of specifically AdaDetect in an adversarial setting.

As far as analysis, in the main text, the authors study two kinds of attacker settings: (1) an Oracle setting and (2) a Practical setting. They present an attack scheme for each setting, and essentially their main theoretical result is an upper bound on the new FDR caused by an oracle attacker.

Experiments conclude the paper. Synthetic data and real-data experiments were conducted.

**Strengths:**

This paper extends and supports conclusions of previous works Zhang et al 2025 and Chen et al 2024. Specifically, the oracle attack schemes that this work outlines can be seen as kind of literal expressions of the p-value perturbation strategy laid out in Chen et al 2024 that sends null p-values into the BH rejection region.

**Weaknesses:**

**Questionable Motivation/Premise and Lack of Novelty:**
-  Considering Chen et al 2024 already established that adversarial perturbations can break the FDR control of current methods leveraging Benjamini Hochberg (BH), I question the level of depth to this work's curiosity surrounding the adversarial robustness of AdaDetect. After all, AdaDetect leverages BH, so, at least at a high level, this fact alone would already suggest there could be vulnerability to adversarial attacks.

- In Marandon et al 2024, AdaDetect roughly has two key elements: (i) conformal p-values and (ii) score functions that do not merely depend on labeled training samples (via one-class classification as in Bates et al 2023). AdaDetect uses conformal p-values so this allows for FDR control via BH, but it also critically departs from methods like Bates et al 2023 by way of its score function (e.g. employing positive-unlabeled learning instead of one-class classification), but this departure is conducted with the aim of enhancing power, not FDR control. Therefore, it's not clear to me why one would figure that AdaDetect somehow has more robust FDR control than any other novelty detection method that uses conformal p-values, including Bates et al 2023. And considering Chen et al 2024 already demonstrated that attacks on methods like Bates et al 2023 on the Credit Card dataset can damage FDR control, the experimental results in this submission on the same Credit Card dataset, only with AdaDetect subbing in now, do not seem like a significant line of study.


**Incomplete theoretical analysis and experiments:**
- The main (and sole) theoretical result is an upper bound on the new FDR that follows after $m_a$ attacks. Specifically, it says that if $\alpha$ is the desired control level and $m_a$- many test-nulls are altered, the new FDR may be no larger than $\alpha + m_a \cdot \mathbb{E}\left[\frac{1}{\tilde{R} \vee 1} \right]$. However, this technically doesn't guarantee that the new FDR will be larger than it would be in the absence of attacks (albeit it seems likely), much less say that the new FDR exceeds $\alpha$ by $m_a \cdot \mathbb{E}\left[\frac{1}{\tilde{R} \vee 1} \right]$. Further, $\tilde{R}$ actually depends on $m_a$, and it's not clear to me that $m_a \cdot \mathbb{E}\left[\frac{1}{\tilde{R} \vee 1} \right]$ is necessarily significant/sizable. Perhaps a lower bound on the new FDR could help in supporting the authors' claims that their Oracle attack scheme critically harms FDR control.
- I think whether or not something exhibits adversarial robustness requires a more nuanced discussion than the authors have provided. For example, if the altered FDR exceeds $\alpha$, but it took a large number $m_a$ of altered samples to achieve this, I'm not sure this would qualify as a lack of robustness. Or, if say, $P_0$ and $P_1$ are supported on rather distinct regions so that nulls and non-nulls are easily distinguishable, then could it not in fact be 'harder' for attacks to be able to reliably succeed in increasing the number of false discoveries - in other words, couldn't there be times when AdaDetect actually does exhibit robustness? Indeed, the authors consider models in which the size of perturbations to samples is immaterial, only the number of perturbations, which mirrors the adversarial setup in Chen et al 2024, in which it is explained that the number of perturbed samples required to alter FDR actually can be made greater in certain settings. For example, the authors could vary the size of the $\mu$ vector that separates their Independent Gaussians in their synthetic data experiments. The authors can also toggle the $\delta$ (if I'm not mistaken, the authors don't report what $\delta$ was set to) parameter that goes into separating their Exchangeable Gaussians. Could they possibly find, as in Chen et al 2024, that when the null and non-null distributions are ``far" from each other that it takes larger $m_a$ to alter the FDR appreciably? These are just a few questions worth investigating for a paper studying adversarial robustness.

**Attack Model**
- The `practical scenario' modeled in Section 3.3, in which the attacker has access to a user's AdaDetect machinery (i.e. can query it) reads a bit odd to me. Would the authors kindly help motivate this setting for me with an example? Furthermore, in this setting, the authors assume that the user's AdaDetect has 'reasonable' detection power, and there is something about an attack intensity $\gamma$ parameter that is not explained?

- The authors' claim more realism than previous works because theirs studies how to directly attack data whereas previous papers consider attacks on the p-values post-transformation of the data. Firstly, I think this marketed claim is a bit misguided; after all, the authors' goal surely couldn't be to provide a realistic recipe for an unrealistic Oracle to go surgically attack the data that is to be fed to AdaDetect? I appreciate that this work and its predecessors reveal potential fragilities of novelty detection methods that use BH, and for such analysis, the language of p-values can be considered as a useful standardization, rather than a shortcoming that requires more realism. Secondly, whether its p-values or data that is being attacked, the idea is the same: make test nulls be mistaken for non-nulls by the BH method step of AdaDetect, which in this paper is achieved by ultimately perturbing test-null data to cross a decision boundary of the $g$ function, but this is equivalent to altering the position of a test null's p-value to be sufficiently small so that it crosses into the BH's rejection region. This correspondence between perturbation of test-null data and perturbation of its would-be p-values may be rough, but if we recall that BH's rejection region is a function of only relative positions of the p-values, an exact mapping between these kinds of perturbations does not seem critical.

**Writing, typos, minor comments, etc.**
- The paper could use a fair bit of revising. I list some examples below.
- Line 123-126: What is the point to the introduction of $X^{train}$ and $X^{test}$ notation? It would seem like it doesn't get used $\ldots$
- In Assumption 1, I think $\pi$ should actually be taken as any permutation of the set $\{1, \ldots, n\} \cup \{ n+ i: i \in \mathcal{H}_0\}$
- Theorem 1: I think it's better if $\tilde{R}$ is written $\tilde{R}_{m_a}$ since there is dependency on the attack size $m_a$?
- Line 429 "We apply four mismatched score function configurations" Can you confirm that there is in fact only one mismatched configuration, RF-NN?
- Line 430 "five real-world datasets" Can you confirm there are in fact only 4 real-world datasets that were used?
- Line 434 It's perhaps a bit surprising as well as apparently unsubstantiated that using a NN configuration as the attacker's model can substantially speed up the attack process. I don't believe I see any clocked times being reported.

**Questions:**

- In light of Chen et al 2024's study on adversarial perturbations threatening the FDR control of novelty detection methods that use BH, including that of Bates et al 2023, I'm wondering what the authors found particularly interesting about pondering the effects of attacking AdaDetect (which uses BH)? Why did the authors find specifically AdaDetect to be such a worthy algorithm or poster child? Was there some kind of feature to AdaDetect that led the authors to conjecture that it might have more robust FDR control over other novelty detection methods that use BH?

- In the Oracle attack scheme, can we not forget about training a score function $g(z)$ and instead replace it with equation (2)'s $s$ score function? If so, if I'm understanding correctly, it seems odd that an oracle attacker ever risks having a mismatched configuration, as was explored in Experiment 3.

- With regards to Theorem 1,...could there be a lower bound on how much the oracle attack scheme increases FDR?

- Could there be regimes where AdaDetect IS robust to adversarial attacks? For example, the authors could vary the size of the $\mu$ vector that separates their Independent Gaussians in their synthetic data experiments. The authors can also toggle the $\delta$ (if I'm not mistaken, the authors don't report what $\delta$ was set to) parameter that goes into separating their Exchangeable Gaussians. Perhaps the authors will find, as in Chen et al 2024, that when the null and non-null distributions are ``far" from each other that it takes larger $m_a$ to alter the FDR appreciably? On a related note, was there some kind of methodology behind the choices of $m_a = 200$ and $m_a = 50$?

- The `practical scenario' modeled in Section 3.3, in which the attacker has access to a user's AdaDetect machinery reads a bit odd to me. Could the authors help motivate this setting for me with an example? Furthermore, in this setting, the authors assume that the user's AdaDetect has 'reasonable' detection power, and there is something about an attack intensity $\gamma$ parameter that is not explained/elaborated?

---

### Official Review · Reviewer_4acG · 2025-11-04

**Soundness:** 3
**Presentation:** 3
**Contribution:** 2
**Rating:** 4
**Confidence:** 3

**Summary:**

This paper studies the behaviour of an existing anomaly detection algorithm `AdaDetect'  under adversarial robustness using two existing black-box adversarial algorithms.  An effective attack scheme is proposed that only requires query access to AdaDetect’s output labels. The vulnerability of AdaDetect is evaluated on synthetic and real-world datasets.

**Strengths:**

The paper presents a lot of theoretical analysis of the behaviour of AdaDetect using two attack methods.

**Weaknesses:**

1. This paper primarily presents an analysis of an existing method, AdaDetect, using two established adversarial attack algorithms. This focus considerably limits the scope of the work. While AdaDetect itself may be a well-designed approach, it is not widely adopted or recognised in the community.

2. The main contribution appears to be the application of these attack algorithms to AdaDetect in a novel manner; however, the level of novelty is rather limited. The study correctly identifies that adversarial perturbations increase the false positive rate of AdaDetect, but it does not propose or explore any solutions to enhance the model’s robustness against such attacks.

3. Furthermore, the experimental analysis is confined to a single detection method. It remains unclear how the insights gained from AdaDetect are generalised to all error-controlled novelty detection methods.

4. A broader evaluation would significantly strengthen the work.

**Questions:**

Please respond to the queries in the weakness section.

---

### Note · Authors · 2025-12-01

I have read and agree with the venue's withdrawal policy on behalf of myself and my co-authors.